# Metallothioneins, a Part of the Retinal Endogenous Protective System in Various Ocular Diseases

**DOI:** 10.3390/antiox12061251

**Published:** 2023-06-10

**Authors:** Daniel Jamrozik, Radosław Dutczak, Joanna Machowicz, Alicja Wojtyniak, Adrian Smędowski, Marita Pietrucha-Dutczak

**Affiliations:** 1Department of Physiology, Faculty of Medical Sciences in Katowice, Medical University of Silesia, Medyków 18, 40-752 Katowice, Poland; jamrozik13@gmail.com (D.J.); pecetomat@gmail.com (R.D.); d201120@365.sum.edu.pl (J.M.); dopracyinauki@gmail.com (A.W.); asmedowski@sum.edu.pl (A.S.); 2GlaucoTech Co., Gen., Władysława Sikorskiego 45/177, 40-282 Katowice, Poland

**Keywords:** metallothionein, ocular diseases, AMD, diabetic retinopathy, glaucoma, cataracts, retinitis pigmentosa, retinal ganglion cell, retina

## Abstract

Metallothioneins are the metal-rich proteins that play important roles in metal homeostasis and detoxification. Moreover, these proteins protect cells against oxidative stress, inhibit proapoptotic mechanisms and enhance cell differentiation and survival. Furthermore, MTs, mainly MT-1/2 and MT-3, play a vital role in protecting the neuronal retinal cells in the eye. Expression disorders of these proteins may be responsible for the development of various age-related eye diseases, including glaucoma, age-related macular degeneration, diabetic retinopathy and retinitis pigmentosa. In this review, we focused on the literature reports suggesting that these proteins may be a key component of the endogenous protection system of the retinal neurons, and, when the expression of MTs is disrupted, this system becomes inefficient. Moreover, we described the location of different MT isoforms in ocular tissues. Then we discussed the changes in MT subtypes’ expression in the context of the common eye diseases. Finally, we highlighted the possibility of the use of MTs as biomarkers for cancer diagnosis.

## 1. Introduction

### 1.1. Structure, Expression and Isoforms of Metallothioneins (MTs)

Metallothioneins (MTs) are small-molecular weight (6–7 kDa) cysteine and metal-rich proteins characterized by their highly conserved structure. MTs bind monovalent or divalent metals, and human MTs consist of 61–68 amino acid residues (cysteine residues make up 25–30%). MTs are composed of two domains: α and β. The α domain (which contains 11 cysteine residues) binds four Zn^2+^, four Cd^2+^ or six Cu^+^ ions, while the β domain (which contains nine cysteine residues) binds three Cd^2+^, three Zn^2+^ or six Cu^+^ ions [1,2,3]. Furthermore, other heavy metals, such as mercury (Hg), may also be sequestered by these proteins [4]. These unique proteins were characterized over 40 years ago by Margoshes and Vallee and were isolated from horse kidney [5,6]. In humans, four isoforms of MTs have been identified: MT-1, MT-2 (also known as MT-2a), MT-3 and MT-4. At least 10 out of 17 MT genes, located on chromosome 16, encode multiple isoforms of MT-1 (MT-1A, -B, -E, -F, -G, -H, -M, -X) and one isoform of MT-2a. Single genes code for MT-3 and MT-4 [1,6,7]. MT-1 and -2 have high sequence homology and are often described together as MT-1/2. Moreover, MT-1 and -2 are the most widely expressed isoforms in the brain (primarily expressed by astrocytes), liver and kidney, and they are highly induced by a variety of stimuli, including metals, hormones, cytokines, growth factors, oxidants, stress and irradiation [2,8,9,10]. MT-3 is expressed mainly in the brain (neurons, astrocytes, cortex, hippocampus) but also in the heart and kidney, while MT-4 is found in the squamous epithelia (mouth, upper gastrointestinal tract and skin) and maternal deciduum (Figure 1) [1,6]. MT-3 and -4 are constitutively expressed despite signal changes in vitro or in vivo [10,11].

### 1.2. Functions of MTs

The well-known biological functions of MTs are related to their high affinity for heavy metals. MTs play an important role in metal homeostasis, especially in zinc (Zn)/copper (Cu) balance and detoxification (through binding the heavy metals). The high affinity of the thiol groups (SH) of MTs to Zn and possibility of passing this metal to other Zn-binding proteins indicates that these proteins can be crucial in the regulation of gene expression. DNA-binding proteins with zinc finger domains control DNA transcription processes. It is known that transcription factors such as the p53 protein, nuclear factor-𝜅B (NF-𝜅B), specificity protein 1 (Sp1) and transcription factor IIIA (TFIIIA) can interact with MTs and change their function. The central domain of p53 includes a zinc finger motif, and zinc is required for its structural stabilization. Zinc-free MT, known as apothionein (apo-MT) or thionein, can remove Zn from p53 and decrease its transcriptional activity with no ability to bind DNA [12,13,14,15]. The same phenomenon takes place in the case of MT interaction with the p50 subunit of NF-𝜅B, where MT is involved in the stabilization of the complex p50 with DNA [16,17]. Moreover, for other transcription factors—Sp1 and TFIIIA—such cooperation has been widely documented [18,19]. It must be highlighted that MTs, as an intracellular source of Zn, participate in the catalytic activation of metalloenzymes and maintain their structural integrity. Zn ensures the structural stabilization of nitric oxide synthase (NOS), matrix metalloproteinase-9 (MMP-9), *δ*-aminolevulinic dehydrase (ALAD) and superoxide dismutase (Cu/Zn SOD) [20,21,22,23,24]. Interestingly, because of their high cysteine content, MTs are able to readily sequester reactive oxygen species and protect cells against oxidative stress. It is suggested that these proteins, as a secondary antioxidant, cooperate with glutathione in maintaining the cellular redox state. MTs become active, especially when the reduced form of glutathione (GSH) is blocked, and act as effective scavengers of free radicals through the Zn–MT redox system. This redox cycle is illustrated in Figure 2. In the cycle, Zn is released from MT during the oxidation of SH groups in MT molecules. As a result of this process, MT-disulfide (thionin) is created. When the oxidized environment becomes reduced, due to, for example, an increase in the glutathione/glutathione disulfide ratio (GSH/GSSG), MT-disulfide is reduced to MT-thiol (thionein). This process is accelerated in the presence of a selenium-derived catalyst, and a further reduced form of MT is able to bind Zn [1,25,26,27,28]. It is interesting to note that the ratio of GSH/GSSG is crucial for a redox cycle of MT and for the amount of Zn released from MT. A relatively high concentration of GSH inhibits the release of Zn and stabilizes the structure of MT. The opposite phenomenon was observed in the presence of GSSG when Zn was realized from MT. Thus, the state of cellular GSH/GSSG determines the quantity of this metal in the cell and controls its transfer to other proteins, such as metalloenzymes or transcription factors. Curiously, it is estimated that MTs are 50 times more efficient as free radical scavengers than reduced glutathione [29]. It is interesting to note that MTs are among the most well-documented factors showing protective effects following brain and peripheral nerve injury (both mechanical injury and ischemia) and in neurodegenerative diseases [2,9,30,31,32,33,34,35,36]. These proteins inhibit proapoptotic mechanisms and enhance the cell differentiation and survival, including neuroprotection and tissue regeneration [37]. The immunomodulatory actions of MTs are related to the direct inhibition of white blood cells, including leukocytes and macrophages, thus reducing the inflammatory response [38]. MTs can also secondarily enhance the expression of important growth factors, such as the brain-derived neurotrophic factor (BDNF), nerve growth factor (NGF), neurotrophins (NTs), fibroblast growth factor (FGF), transforming growth factor-beta (TGF-β), vascular endothelial growth factor (VEGF) and their receptors [38,39]. Recently, many studies have suggested that MTs may play a vital role in carcinogenesis and provide potential promising markers for cancer (Figure 3) [10,40,41,42,43].

### 1.3. Expression of MTs in the Ocular Environment (Human Eye)

It has been suggested that, as endogenous antioxidants, MTs play an important role in protecting neuronal retinal cells in the eye. Many reports in the literature have described the expression of MTs in various human ocular tissues (Figure 4). Almost all MT-1 isoforms are abundant in lens and corneal tissue; however, MT-1B and MT-4 are not detected in the ocular environment [44]. Moreover, MT-1/2 is present in the corneal epithelium and endothelial cells, while MT-3 is present in the retinal ganglion cells and the retinal nerve fibre layer [44,45]. Interestingly, Avarez et al. underlined that the relatively higher expression of MT-1/2 isoforms in the anterior segment of the eye (cornea and lens), compared with the posterior segment (retina, RPE (retinal pigment epithelium) and sclera), may reflect their key role as protective mechanisms against oxidative stress and inflammation. Both the cornea and lens, which constitute the first physical barrier, protect the sensitive inner ocular environment from external environmental insults, including ultraviolet radiation. In addition, MT binds six Zn ions and one copper ion under control conditions (Zn6Cu1-MT), whereas pro-inflammatory cytokines or Zn, cause MT to contain seven Zn ions (Zn7-MT). Zn7-MT species have a greater antioxidant capacity than Znx-MT species (where “x” ranges from 0 to 6 Zn atoms). This data suggests that the supplementation of Zn may be important because MTs interact more effectively with reactive oxygen species at a high concentration of Zn than at a low concentration [45]. Oppermann et al. established the expression pattern of MT isoforms in the adult human lens and demonstrated that MT-1E, -F, -G and -H isoforms were present in both the lens epithelium and fibre cells. However, the expression of MT-1E and F isoforms was higher in the epithelium than in the lens fibres. MT-2A was detected manly in the lens epithelium [46]. Using an RT-PCR analysis and in situ hybridization, Tate et al. showed that the MT isoforms MT-1/2 and -3 were localized mainly in the retinal pigment epithelium (RPE) and the photoreceptor layer of the retina. The authors suggested that the localization of MTs and the response of these proteins to oxidative stress were consistent with MTs’ role as an antioxidant in the RPE and retina [47]. Therefore, the prevalence of MTs in different parts of eye and their higher expression after exposure to oxidative stress caused by daily exposure to ultraviolet light or chemical insults indicates their key role in defence against oxidative damage. Oxidative damage and inflammatory processes are the main reasons behind various age-related eye diseases, including glaucoma, age-related macular degeneration (AMD), diabetic retinopathy and retinitis pigmentosa (RP) [3,48,49,50,51]. Moreover, mutations of genes encoding antioxidant enzymes may lead to a faster progression of these diseases or cause inherited retinal dystrophies (IRDs) [52]. More than 250 genes causing IRDs have been identified using next-generation sequencing (NGS) techniques [53]. About 70 genes have already been associated with RP, the most common being IRD [54]. Among them, five genes, namely kelch-like protein 7 (KLHL7), retinol dehydrogenase 11 (RDH11), ceramide kinase-like (CERKL), aryl hydrocarbon receptor interacting protein like 1 (AIPL1) and usher syndrome type 1G (USH1G), have been considered to have a close connection between induced oxidative stress and RP onset and progression [54,55].

## 2. MTs in Ocular Diseases

### 2.1. Age-Related Macular Degeneration (AMD)

AMD is one of the main causes of visual impairment and severe vision loss. AMD is classified into early and late stages, with dry (atrophic) and wet (neovascular) forms. The early stage of AMD is characterized by clinical signs such as drusen and abnormalities of the RPE, which contribute to the dysfunction and loss of RPE, leading to photoreceptor death. RPE degeneration damages Bruch’s membrane and upregulates the VEGF, which promotes the outgrowth of abnormal choroidal vessels underneath the RPE and retina, and, finally, the late stage of AMD is developed. Age, genetic predisposition, smoking, lifestyle, diet and nutrition are among the most well-documented risk factors for AMD. These various risk factors are responsible for oxidative damage and inflammatory-mediated processes. It is generally accepted that the production of reactive oxygen species increases in patients with AMD. Golestaneh et al. reported that RPE obtained from AMD donors showed increased levels of ROS and greater sensitivity to oxidative stress than RPE obtained from non-AMD donors [56]. The RPE regulates the oxidative stress using protective mechanisms, which include antioxidant enzymes, such as superoxide dismutases (e.g., cytosolic copper-Zn superoxide dismutase or mitochondrial manganese superoxide dismutase), molecular chaperons, such as heat shock proteins (HSPs) and the ubiquitin-proteasome pathway, which participates in refolding or degrading damaged proteins [57].

Previous studies have suggested that a diet supplemented with antioxidants, such as vitamins C and E, carotenoids and Zn, delays the progression of AMD [58,59,60,61]. Zn-enriched diets are of particular interest; some studies have shown that the content of Zn in human RPE/choroid in AMD was decreased by 24% and that an exceptionally high level of Zn was observed in drusen [62]. Therefore, because MTs play pivotal roles in Zn homeostasis and the Zn–MT complex neutralizes free radicals, it seems that MTs can affect drusen formation during AMD. Ageing and oxidative stress decrease the amount of MTs in the macula and trigger the release of Zn from MTs into the extracellular space, which can promote the formation of drusen [44]. It is important to note that a Zn-deficient diet reduces the expression of MT in both the RPE and the retina, increasing the levels of retina lipid peroxidation [63]. The results of the AREDS1 trial, in which patients took Zn supplements, seem to be promising, as the progression rate from early to late AMD after six years of supplementation was delayed [64]. Wang et al. showed that a small antioxidative molecule, D609, can be used as a potential treatment for AMD due to its properties of efficiently inhibiting oxidative damage-induced RPE cell death both in vitro and in vivo. The authors also noted that D609 has the unique ability to upregulate the expression of MT-1E, -G, -X isoforms and MT-2A. Moreover, when MT expression was silenced by siRNA interference, the protective effect of D609 was significantly decreased, indicating the essential role of MT in the antioxidative effects of D609 [65]. Ahmen et al. also observed the increase in the level of MT-1 mRNA when testing the protective properties of xaliproden (5-OH-tryptamine 1a receptor agonist) for RPE cell damage in AMD [66,67].

### 2.2. Retinitis Pigmentosa (RP)

RP is caused by mutations in primarily rod-specific genes, leading to a progressive and irreversible loss of photoreceptor cells. At the more advanced stage of the disease cone cells also die, which can ultimately cause blindness. One of the first symptoms reported by patients is night blindness, followed by a loss of various peripheral and central vision functions and colour vision defects. Of RP cases, 50–60% represent the autosomal recessive trait, 30–40% are autosomal dominant and 5–15% are X-linked RP [68,69]. Wunderlich et al. analysed the expression of MT-1/2 in three rodent models of RP and confirmed that MTs were expressed in mouse and rat retinas and the pigment epithelium. Moreover, the expression of these proteins correlated with glial activation (mainly the Müller cells) and the megalin receptor was expressed both in the inner and outer retina. They suggested, based on the progressive loss of megalin, that the transport of MT-1/2 from glial cells into the degenerating photoreceptors could be impaired, limiting the actions of MTs. Furthermore, they pointed out that the expression of these MTs was tightly regulated by the levels of growth factors and cytokines [70].

### 2.3. Diabetic Retinopathy (DR)

DR is one of the most common complications of diabetes leading to vision impairment. The prevalence of DR is higher in patients with type 1 diabetes than in those with type 2 and reaches about 35.4% [71]. Several cellular functions, such as intracellular calcium level, NADPH oxidase activity and the signalling of nuclear factor kappa-light-chain-enhancer of activated B cells (NF-*κ*B), are altered by the high glucose level [72,73]. Furthermore, the main cause of oxidative stress, hyperglycaemia, stimulates the production of free radicals and reactive oxygen species (ROS). Oxidative stress promotes the release of inflammatory cytokines, such as interleukin-6 (IL-6), tumour necrosis factor-α (TNF-α) and interleukin 1ß (IL-1ß), which lead to retinal vasculature damage [74,75]. Alterations in the integrity of retinal capillaries and their occlusion, vascular leakage, subsequent neovascularization and retinal haemorrhages are the typical symptoms of vascular pathology in DR. As it progresses, compromised retinal microvascular circulation causes ischemia, which increases the size and number of intraretinal haemorrhages, and, finally, cotton wool spots may also appear as a sign of neuronal component damage. At every stage, macular oedema may occur as a result of the breakdown of the blood–retinal barrier [76]. It is well known that hypoxia and oxidative stress increase VEGF production; its upregulation has been very well documented in the early stage of diabetes in numerous studies [77,78,79]. Some studies have found evidence of retinal neurodegeneration occurring before the onset of vascular alterations [77,80,81,82,83]. Nakamura et al. drew attention to the correlation between MT-1/2 and retinal neovascularization and reported that the levels of MT-1/2 and VEGF in the vitreous fluids were significantly higher in proliferative diabetic retinopathy (PDR) patients than in control patients. Moreover, the ocular neovascularization in MT-1/2 knockout (KO) mice was decreased via the downregulation of hypoxia-inducible factor (HIF-1α) and this in vitro study demonstrated that MT-1/2 promoted the ubiquitination of HIF-1α [84]. Other reports suggest that MTs can supply Zn, which actively regulates insulin function. It is generally accepted that Zn is required in pancreatic β-cells in the process of insulin biosynthesis and the maturation of insulin secretory granules; thus, changes in Zn levels in the pancreas are associated with diabetes [85,86,87]. Furthermore, it is suggested that Zn, through the inhibition of NADPH oxidase, can prevent retinal pericyte apoptosis in DR [88,89]. Moustafa underlined that Zn protects the retina from diabetes-induced increased lipid peroxidation and decreased glutathione levels, mainly through inducing MT synthesis [90]. In addition, nitric oxide (NO) is able to induce Zn release from MTs [91,92]. Proinflammatory cytokines, such as Il-6, Il-1β, TNFα and NO are responsible for the translocation of MT between the cytoplasm and the nucleus [92]. These proinflammatory cytokines are well known as the main cause of retinal vasculature damage [93]. Under normal conditions, MT is located in the cellular cytoplasm; only during proliferation and regeneration is MT found in nuclei. It is suggested that Zn–MT is first translocated to the nucleus and then, as the intracellular NO concentration increases, Zn is released from the nuclear MT. Furthermore, the researchers indicate that NO-induced intracellular Zn release can inhibit further NO production via a feedback loop [6,92].

### 2.4. Glaucoma

Glaucoma is thought to be a neurodegenerative disease that leads to severe visual impairment or vision loss. It is estimated that by 2040 the number of glaucoma patients may reach 110 million [94,95]. Glaucoma is characterized by damage to RGCs and their axons, which leads to visual field loss [39,96]. Elevated intraocular pressure (IOP) is the most common factor responsible for the development of glaucoma [3,97]. However, it is known that high IOP is not the only pathological factor. It must be highlighted that some patients with normal IOP develop typical symptoms of glaucoma (i.e., normal tension glaucoma (NTG), particularly in Asian populations [98]. Therefore, it is suggested that other factors may also be involved in the pathogenesis of this disease, such as a genetic predisposition, ischemia, advanced age, smoking or a deficiency of antioxidants in the diet [91]. The eye antioxidant defence system, including glutathione peroxidase (GPX), superoxidase dismutase (SOD), catalase (CAT) and MTs, is efficient only in physiological conditions. Therefore, the study of the activity of these systems during various stages of glaucoma seems to be a natural direction of research to understand the mechanisms responsible for the progression of this disease. MTs, one of the elements of the eye antioxidant defence system, have aroused the interest of scientists working on the progression of glaucoma.

Suemori et al. demonstrated that MT2 mRNA was upregulated in the injured retina and that MT1/2 knockdown exacerbated retinal damage in a glaucoma model induced by NMDA intravitreal injection [36]. Whereas Hohberger et al. demonstrated that patients with primary open-angle glaucoma and pseudoexfoliation glaucoma had significantly higher aqueous humour levels of Zn. The authors suggested that there is a potential relationship between the trace elements and MT due to the participation of MT in binding these metals and neutralizing free radicals [99]. When investigating the aqueous humour of glaucoma patients, Aranaz et al. noted higher levels of antioxidant capacity and higher concentrations of aqueous and serum magnesium and phosphorus in glaucoma patients. Interestingly, they did not show the significant differences in Cu and Zn composition in glaucoma [90]. However, another report indicated that elevated IOP induced the upregulation of MT and Cu/Zn superoxide dismutase genes in human trabecular meshwork [100]. On the other hand, DeToma et al. showed that the retina of pre-glaucomatous DBA/2J mice had greater Mg, Ca and Zn concentrations than glaucomatous animals. Moreover, the concentrations of these metals were higher in younger animals than in older ones [101].

Our previous study demonstrated that an extract obtained from the predegenerated sciatic nerves (PNE) protects RGCs in a rat glaucoma model. Moreover, we suggested that the key molecule responsible for the beneficial properties of the extract is MT-2. MT-2 was one of the proteins detected in a mass spectrometry analysis of the extract. The screening test using retinal explant culture showed that the density of β3-tubulin positive cells in the ganglion cell layer (GCL) was significantly higher in explants cultured in a standard medium supplemented with MT-2 than in explants in a non-supplemented standard medium. The lowest density of β3-tubulin positive cells was reported in explants cultured in a standard medium supplemented with MT-2 and gentamycin, which serves as a blocker of the megalin receptor linked to MT-2 activity [39]. Furthermore, the lactate dehydrogenase (LDH) levels were decreased; however, in explants treated with MT-2, the decline in LDH activity was significantly slower when compared with that of the non-treated group, suggesting that more cells survived in the treated explants (unpublished data). These results clearly indicate that MT-2 has a neuroprotective effect on RGC. We suspect that MT-2 could induce an increase in endogenous BDNF expression, which we observed in our study [39]. This increased synthesis of BDNF may represent an endogenous neuroprotective response of RGCs, as highlighted by Vecino [102,103].

There are relatively few scientific reports in the field of MTs’ participation in the development of glaucoma, and we suggest that further research should be undertaken in this area.

### 2.5. Cataracts

Cataracts are another ocular disease involving the loss of lens clarity, which can lead to severe visual impairment. Cataracts can have various causes, including developmental abnormalities, trauma, metabolic disorders and drug-induced changes, but ageing seems to be the most common cause of the development of this disease. Moreover, many studies have suggested that women are at slightly greater risk of cataract development than men [104,105,106]. Kantorow et al. provided evidence that age-related cataracts are associated with alterations in the expression of multiple epithelial genes, including MT-2. The authors showed the overexpression of MT-2 and the underexpression of the protein phosphatase 2A regulatory subunit (P2A-RS) in cataract epithelia, underlying that these proteins maintain lens transparency [107]. Hawse’s research team indicated that the overexpression of MT-2A protected human lens epithelial cells against oxidative stress induced by cadmium and tertiary butyl hydroperoxide (TBHP), suggesting that MT-2A may play a role in regulating the expression of other important antioxidant genes in response to oxidative stress [108]. Oppermann, defining the spatial expression patterns of MTs in lens epithelia and fibres, demonstrated that MT-2A and five isoforms of MT-1 (-E, -F, -G, -H, -L) were detected almost exclusively in the lens epithelium, whereas the isoforms of MT-1 were present at high levels in both lens epithelia and fibres [46].

## 3. Future Perspectives

MTs play a pivotal role in multiple biological processes, including metal ion homeostasis and detoxification; the regulation of cell growth and proliferation; protection from oxidative stress; and negative processes, such as tumour growth, differentiation, angiogenesis and drug resistance. For this reason, MTs seem to be potential therapeutic targets and promising markers for cancer. Recently, the participation of these proteins in carcinogenesis, wherein the direction of changes in MT expression depends on the type of cancer and its location, has aroused great interest. MT expression is upregulated in breast cancer, nasopharyngeal cancer, ovarian cancer, urinary bladder cancer and melanoma, but in prostate cancer, papillary thyroid carcinoma or hepatocellular carcinoma, the expression of these proteins is downregulated [10]. Future work should focus on understanding the full expression profile of particular MT isoforms in various cancers. This knowledge will enable the development of quick and accurate methods of cancer diagnosis and therapy.

Moreover, many ocular diseases have a genetic basis, and most can be caused by mutations in many different genes. Therefore, next-generation sequencing (NGS) technologies, enabling the study of the mechanisms of inherited retinal diseases, seem to be a very important diagnostic tool for the future. These techniques are currently the gold-standard approach for cost-effective genetic analysis and are also essential for identifying suitable candidates for gene therapies. NGS methods have started a real revolution in the biomedical sciences.

## Figures and Tables

**Figure 1 antioxidants-12-01251-f001:**
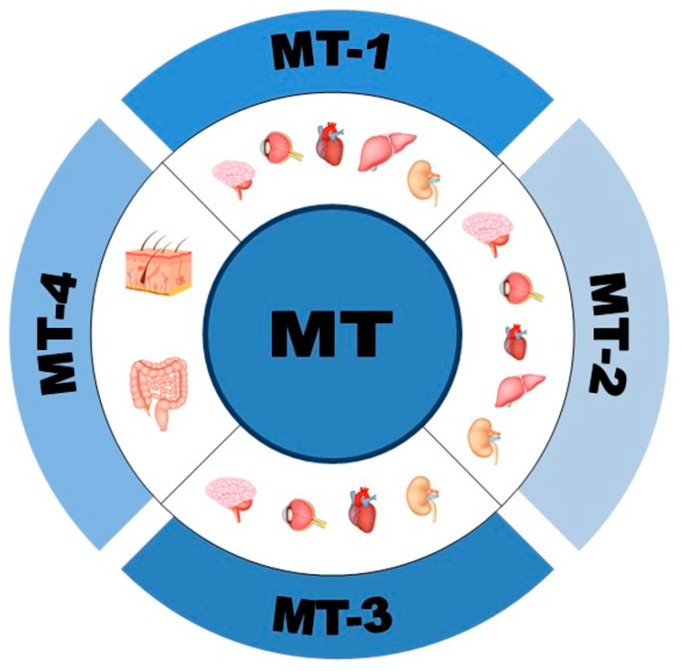
The most common location of metallothionein isoforms in the organs.

**Figure 2 antioxidants-12-01251-f002:**
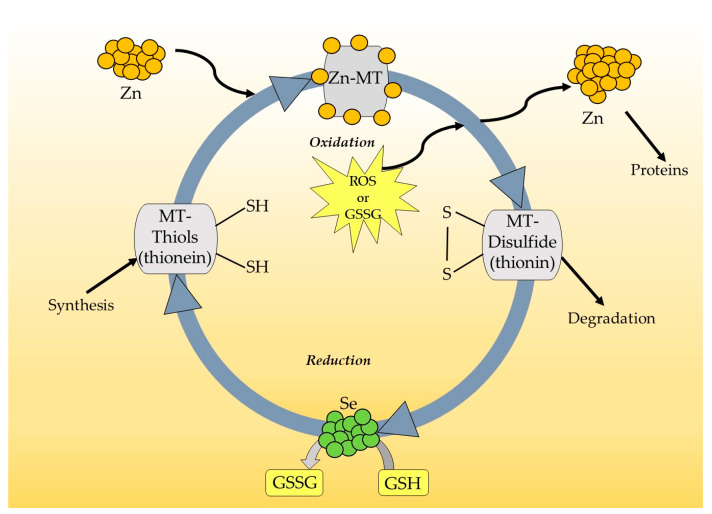
Metallothionein redox cycle.

**Figure 3 antioxidants-12-01251-f003:**
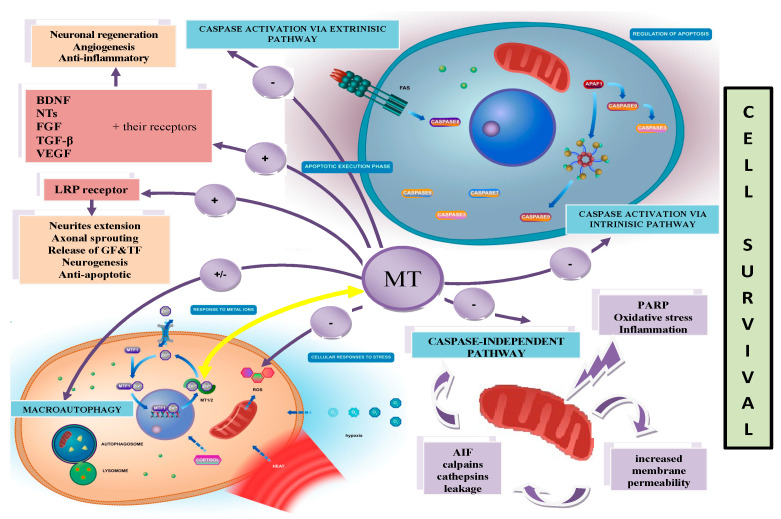
Functions of metallothioneins.

**Figure 4 antioxidants-12-01251-f004:**
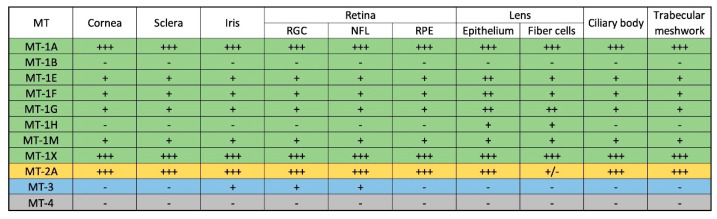
The expression of different metallothionein isoforms in the ocular tissues (+++ high; ++ medium; + low; +/− present only in selected samples; − absent; RGC—retinal ganglion cells; NFL—nerve fibre layer; RPE—retinal pigment epithelium) [44,45,46,47].

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
