# Peer review of "Metallothioneins, a Part of the Retinal Endogenous Protective System in Various Ocular Diseases"

_antioxidants, 2023, doi:10.3390/antiox12061251_

Round 1

Reviewer 1 Report

Jamrozik et al. realized a very interesting review describing the “Metallothioneins, a part of retinal endogenous protective system in various ocular diseases”. I consider the manuscript very interesting but, at the same time, I suggest several revisions needed to improve the reliability and the completeness of the paper: 

I suggest adding data related to recent approaches to identify novel mutations and related alteration of molecular pathways in retinal dystrophies patients by using innovative NGS analyses pipelines, especially focusing on oxidative stress. 

The “Future perspectives” section should be improved, also referring to novel technologies and recent genome editing techniques. 

Manuscript requires important English revisions and typos correction. 

Author Response

I attached pdf

Reviewer 2 Report

The ms quoted "antioxidants-2392921" and titled "Metallothioneins, a part of retinal endogenous protective 2 system in various ocular diseases" describes metallothioneins as important proteins in retinal departments.

This review focused its attention on the therapeutic aspects associated with specific diseases.

I suggest to include a paragraph in which modulators are described, divided into categories. 

Author Response

Responses to Reviewer 2 comments
It is extremely kind of you to take the time to review our text and write the comments on the manuscript.
Responses for specific comments:
Point 1. We reviewed all reports regarding metallothioneins very precisely one more time. We found only two drugs affecting metallothioneins expression: D609 and xaliproden. Results of these reports were included in Section 2.1 (AMD). However, we decided to enrich this section with more detailed data - we added information about specific MT isoforms, which expression increased after application of these drugs. Due to the small number of reports, it seems that inserting the description of these drugs in section about AMD will be more convenient for audience. MTs are not treated as standard markers of oxidative stress - researchers usually choose other markers, that’s why there’s so few reports.

Reviewer 3 Report

The current manuscript aims to describe a comprehensive review on the topic of metallothioneins as a part of retinal endogenous protective system in various ocular diseases. In particular, the authors attempt to summarize the relationship between metallothioneins and various ocular diseases. Overall, in my opinion, this brief review-type article is significant in the area of neuroprotective antioxidants. Therefore, the manuscript may be suitable for publication to this journal if the authors’ improvements are deemed adequate.

Specific comments:

1.         Over the past few years, some review papers on the same topic have been published in the literature database. Please refer to the following papers: #1 DOI: 10.1248/bpb.b17-00856 [also reference 3] & #2 DOI: 10.3390/antiox10010089 [also reference 27]. The authors should carefully clarify the differences in the academic contribution points between the current manuscript and those earlier reports.

2.         As mentioned in Section 1.3, MT-1B and MT-4 are not detected in the ocular environment. However, this important claim was not supported by any documented reference. Please improve.

3.         Given that this review focuses on the MTs in ocular diseases, the authors should clearly indicate the MT subtypes in each individual ocular tissue.

4.         The term “oxydation” in Figure 2 is not expressed in English. Please check and make necessary change.

5.         The authors described MTs in ocular diseases in Section 2. Although many examples were introduced, no illustration was given to enrich the article content and to attract the attention from the readers. Please further improve.

6.         In Section 3, the authors described the findings of their work. But, such manuscript structure is rarely seen in the manuscript writing in the review-type article. In particular, the relationship between MTs and glaucoma has been reported in Section 2.4. The authors should consider the rewriting of this section by incorporating with others’ findings to balance scientific viewpoints.

7.         Furthermore, as mentioned in Section 2.4, there is relatively little research in this field, and we suggest that further research should be undertaken in the areas describing in the following sections. But, the audiences are unaware of the meaning of this sentence. Please further specify.

8.         In Section 2.4, the authors described the literature reports about the metal composition in aqueous humor of glaucoma patients. However, they did not give any relevant details regarding the metal composition in the posterior segment of the eye. Please improve.

9.         The authors provided the detailed descriptions in Section 2.4. But, the content of glaucoma should be more concise to avoid the loss of focus. Please improve.

10.      Where is Section 5? Please check the section heading and make necessary correction again.

11.      As mentioned in the Abstract Section, our previous study demonstrated using a rat glaucoma model that MT-2 is included among the extracts from the sciatic nerves with high neurotrophic potential towards retinal ganglion cells (RGC). Moreover, the density of β3tubulin positive cells in the ganglion cell layer was significantly higher in explants cultured in standard medium supplemented with MT-2 than in explants in non-supplemented standard medium. These results clearly indicate that MT-2 has a neuroprotective effect on RGC by the induction of endogenous brain-derived neurotrophic factor (BDNF) expression. However, the audiences are unaware of the underlying reason of emphasizing the description of previous work in this important section. Please revise the writing of Abstract since this is one of the most important core paragraph in the manuscript.

12.      As stated by the authors, these proinflammatory cytokines are well known as the main cause of retinal vasculature damage. However, this important claim was not supported by any documented reference. The authors are highly recommended to consider the inclusion of the following relevant paper involving retinal antioxidant defense system and anti-inflammatory/anti-angiogenic activities (please refer to DOI: 10.1021/acsnano.2c05824) in the reference list to balance scientific viewpoint and update the article content.

The quality of English in this paper is acceptable.

Round 2

Reviewer 1 Report

Manuscript can be accepted in present form

Reviewer 3 Report

The authors have made significant changes in the revised version. Following appropriate improvements, the manuscript is now suitable for publication in this high-quality journal.